# Inhibiting F-Actin Polymerization Impairs the Internalization of *Moraxella catarrhalis*

**DOI:** 10.3390/microorganisms12020291

**Published:** 2024-01-30

**Authors:** Jinhan Yu, Jingjing Huang, Rui Ding, Yingchun Xu, Yali Liu

**Affiliations:** 1Department of Clinical Laboratory, State Key Laboratory of Complex Severe and Rare Diseases, Peking Union Medical College Hospital, Chinese Academy of Medical Sciences and Peking Union Medical College, Beijing 100730, China; pumc_yujinhan@student.pumc.edu.cn (J.Y.); hjj_6620@163.com (J.H.); 2Graduate School, Chinese Academy of Medical Sciences and Peking Union Medical College, Beijing 100730, China; 3Beijing Key Laboratory for Mechanisms Research and Precision Diagnosis of Invasive Fungal Diseases, Beijing 100730, China; drdingrui@163.com

**Keywords:** *Moraxella catarrhalis*, cytoskeleton, F-actin, polymerization, respiratory epithelial cells, chronic obstructive pulmonary disease

## Abstract

*Moraxella catarrhalis*, a commensal in the human nasopharynx, plays a significant role in the acute exacerbation of chronic obstructive pulmonary disease (AECOPD). Its pathogenicity involves adherence to respiratory epithelial cells, leading to infection through a macropinocytosis-like mechanism. Previous investigations highlighted the diverse abilities of *M. catarrhalis* isolates with different phenotypes to adhere to and invade respiratory epithelial cells. This study used a murine COPD model and in vitro experiments to explore the factors influencing the pathogenicity of distinct phenotypes of *M. catarrhalis*. Transcriptome sequencing suggested a potential association between actin cytoskeleton regulation and the infection of lung epithelial cells by *M. catarrhalis* with different phenotypes. Electron microscopy and Western blot analyses revealed a decrease in filamentous actin (F-actin) expression upon infection with various *M. catarrhalis* phenotypes. Inhibition of actin polymerization indicated the involvement of F-actin dynamics in *M. catarrhalis* internalization, distinguishing it from the adhesion process. Notably, hindering F-actin polymerization impaired the internalization of *M. catarrhalis*. These findings contribute vital theoretical insights for developing preventive strategies and individualized clinical treatments for AECOPD patients infected with *M. catarrhalis*. The study underscores the importance of understanding the nuanced interactions between *M. catarrhalis* phenotypes and host lung epithelial cells, offering valuable implications for the management of AECOPD infections.

## 1. Introduction

*Moraxella catarrhalis*, a human nasopharyngeal commensal, is occasionally isolated from patients with otitis media and chronic obstructive pulmonary disease (COPD) [1]. Additionally, it is the predominant etiological pathogen of acute exacerbation of COPD (AECOPD) [2,3]. As the condition worsens, patients with COPD experience more acute exacerbations, leading to a gradual decline in lung function and increased death risk [4,5,6]. However, given its similarity to the commensal *Neisseria* species in culture, *M. catarrhalis* is often overlooked in human respiratory system samples [3]. Nevertheless, *M. catarrhalis* is detected in 10–20% of bacterial cultures of samples from patients with AECOPD [7]. In fact, it accounts for 2–4 million adult AECOPD cases each year in the USA, leading to 700,000 hospitalization events [3,8].

With the extensive use of antibiotics and the introduction of vaccination programs against pneumococci, the multidrug resistance of *M. catarrhalis* has gradually increased, particularly to combined β-lactams and macrolides. The positivity rate for *M. catarrhalis* β-lactamase is >95%, with no noticeable regional differences [9,10,11,12]. Meanwhile, macrolide-resistant *M. catarrhalis* has prominent regional characteristics in China [13], with the highest macrolide resistance or insensitivity reported in Beijing (65–85%) [10,11]. Previously, we reported that macrolide-resistant isolates are highly concentrated in the CC449 (CCN10) and CC363 clonal complexes, whereas macrolide-susceptible isolates are concentrated in the CC224 and CC446 (CCN08) clonal complexes [13,14]. Indeed, the macrolide susceptibility of *M. catarrhalis* is correlated with its ability to adhere to and invade human respiratory epithelial cells [14]. For instance, the highly resistant clonal complex group, CC363, exhibits a strong ability to adhere to A549 cells. In contrast, all CC446 clonal complexes are macrolide sensitive, and they exhibit a strong ability to invade A549 cells.

During the pathogenesis of *M. catarrhalis*, it needs to interact with the host in order to elicit an immune response through various extracellular and intracellular receptors. It can adhere to, colonize, and subsequently infect respiratory epithelial cells via a macropinocytosis-like mechanism and then interacts with intracellular receptors, such as nucleotide-binding oligomerization domain 1 (NOD1), activating downstream signaling pathways (such as the NF-κB signaling pathway) to mediate inflammatory responses [15,16]. At present, there are still few studies on how *M. catarrhalis* invades the respiratory epithelial cells of the host.

The interactions between human epithelial cells and respiratory pathogens are highly complex and critical to COPD pathogenesis [17]. Cellular actin exists in filamentous (F-actin) and globular (G-actin) forms, with the filamentous form serving as the major actin cytoskeleton component [18]. However, actin functioning relies primarily on the dynamic polymerization and depolymerization of actin filaments into monomeric G-actin and filamentous F-actin forms [18]. The interactions between lung epithelial cells and *M. catarrhalis* with different phenotypes require further investigation. These studies will provide critical theoretical insights for the prevention and individualized clinical treatment of AECOPD. Accordingly, the aim of this study was to investigate the pathogenic mechanisms of *M. catarrhalis* with different phenotypes.

## 2. Materials and Methods

### 2.1. Isolates

Based on a previous study [14], two *M. catarrhalis* isolates were selected that differed significantly in their ability to adhere to and invade A549 cells (*p* = 0.0286) and in their macrolide susceptibility. The macrolide-resistant isolate 35-OR showed strong adherence to, but weak invasion abilities toward, A549 cells (minimum inhibition concentration >256 µg/mL; percent of invasive/adherent bacteria = 0.000019%). The macrolide-susceptible isolate 73-OR showed weak adherence and strong invasion abilities toward A549 cells (minimum inhibition concentration ≤ 0.25 μg/mL; percent of invasive/adherent bacteria = 0.005170%).

### 2.2. Mouse Model for COPD and Pulmonary Clearance Model

All mice used in this study had a C57BL/6J background. To generate a COPD mouse model, 8–10 week-old male mice were anesthetized with isoflurane and perfused intratracheally with porcine pancreatic elastase (120 U/kg) (Elastin Products Company, Owensville, MO, USA) dissolved in 40 µL of physiological saline solution. Pancreatic elastase was injected seven times, resulting in major emphysematous changes in the lungs after a single intratracheal instillation, as previously reported [19]. This study was approved by the Research Ethics Committee of Peking Union Medical College Hospital (PUMCH, JS-2304, and JS-3452), and the animal study was approved by the Institute of Microbiology, Chinese Academy of Sciences Animal Care and Use Committee (project number: APIMCAS2021144).

As described previously, a mouse pulmonary clearance model was established [14]. C57/BL6J COPD mice were maintained under standard pathogen-free conditions in the animal facility at the Institute of Microbiology, Chinese Academy of Sciences. The mice (*n* = 5–7/group) were anesthetized with isoflurane, and the trachea was surgically exposed. The control (CO), 73-OR-infected, and 35-OR-infected groups were intratracheally injected with phosphate buffer saline (PBS) and 10^8^ colony-forming unit (CFU) of 73-OR and 35-OR *M. catarrhalis*, respectively. At 3 h post-challenge, the mice were sacrificed via CO_2_ exposure.

Peripheral blood and lung tissue samples were collected. The white blood cells (WBCs) were enumerated using an automated hematological analyzer (Sysmex Xs-800i, Norderstedt, Germany). The lungs were homogenized in a tissue homogenizer and plated on blood agar plates (100 µL/plate) containing 5% sheep blood. Following overnight incubation at 37 °C under 5% CO_2_, the CFU was determined via manual counting. Histological analyses were performed as previously described [14]. Briefly, the mice were administered PBS or 108 CFU of *M. catarrhalis* (73-OR or 35-OR) in 50 µL of PBS. Hematoxylin and eosin (H&E) stained pulmonary tissue at 3 h post-infection. Histological changes in the lung tissue sections of each group were compared at ×100 and ×400 magnifications.

### 2.3. RNA Sequencing and Data Analysis

RNA sequencing was performed to explore the interactions between *M. catarrhalis* and A549 cells. A549 cells were seeded in a 24-well cell-culture plate (10^5^ cells/well) and inoculated with *M. catarrhalis* (73-OR or 35-OR isolate at a multiplicity of infection (MOI) of 10) and PBS for 4 h at 37 °C. Total cellular RNA was extracted from the infected cells using TRIzol reagent (Life Technologies, Carlsbad, CA, USA). RNA quality was measured using RNA quality score (RQS), and 3 μg of RNA per sample was used as the input material for RNA sample preparation. Sequencing libraries were generated using the NEB Next^®^ Ultra^®^ RNA Library Prep Kit for Illumina^®^ (#E7530L; NEB, Ipswich, MA, USA) according to the manufacturer’s instructions, and index codes were added to the attribute sequences. The cDNA library preparations were sequenced on the Illumina HiSeq platform to obtain 150 bp paired-end reads.

The mapped reads per gene were counted using HTSeq v0.6.1, and fragments per kilobase of transcript per million mapped reads (FPKM) of each gene were calculated. Log FPKM values were used to calculate Z-scores to generate heat maps using HemI 1.0. *p* adj < 0.05 was used as the threshold to determine the significance of the differences in gene expression. Functional annotation enrichment analysis with Gene Ontology (GO) terms and Kyoto Encyclopedia of Genes and Genomes (KEGG) and Reactome pathways were performed using the Cluster Profiler Bioconductor package (version 2.11). Significance (*p* < 0.05) was determined using Pearson’s χ^2^-test (or Fisher’s exact test, when appropriate).

### 2.4. G-Actin/F-Actin In Vivo Analysis

The G-actin/F-actin In Vivo Assay Biochem Kit (catalog #BK037; Cytoskeleton, Inc., Denver, CO, USA) was used to quantify globular actin (G-actin) and filamentous actin (F-actin). Briefly, the cells were treated with 0.1% dimethyl sulfoxide (DMSO) vehicle control or inoculated with *M. catarrhalis* (73-OR or 35-OR isolate at an MOI of 10). The cell membrane was disrupted, whereas the G- and F-actins remained stable and were maintained by lysing the cells in the lysis and F-actin stabilization buffer. The lysate was then centrifuged at approximately 1000× *g* for 5 min at 25 °C. The supernatant was centrifuged again at 100,000× *g* to separate F-actin (pellet) from soluble G-actin (supernatant). F-actin and G-actin were resolved using 10% sodium dodecyl sulfate-polyacrylamide gel electrophoresis (SDS-PAGE) and transferred to an Immun-Blot polyvinylidene fluoride membrane for Western blot analysis with anti-actin Mab (clone 7A8.2.1). The G-actin/F-actin bands were scanned using densitometry, and the ratio of free G-actin to F-actin was calculated.

### 2.5. Immunofluorescence and Imaging

For actin analysis, A549 cells were inoculated with *M. catarrhalis* at an MOI of 10. The samples were fixed in 3.7% methanol-free formaldehyde solution in PBS for 20 min at 25 °C. After fixation, the samples were permeabilized in PBS containing 0.1% Triton X-100 (PBT) thrice for 10 min each, followed by 1 h of blocking in PBT containing 1% bovine serum albumin and 5% fetal calf serum. The samples were incubated for 15 min with 4′,6-diamidino-2-phenylindole (Hoechst 33342) in PBS. F-Actin filaments were labeled with fluorescein isothiocyanate fluorescent-conjugated phalloidin, Alexa Fluor 488 phalloidin (A12379; Invitrogen, Waltham, MA, USA) for 1 h at room temperature.

The samples were mounted in VECTASHIELD Antifade Mounting Medium (H-1000; VECTOR Laboratories, Newark, CA, USA). Imaging and analysis were performed using a high-content imaging device (Operetta CLS; PerkinElmer, Hamburg, Germany) and its associated software (Harmony 4.9; PerkinElmer, Hamburg, Germany).

### 2.6. Actin Polymerization Inhibition Assay

Cytochalasin D (Zygosporin A) inhibits G-actin–cofilin interactions by binding to G-actin. Cytochalasin D also inhibits cofilin binding to F-actin and decreases actin polymerization and depolymerization rates in live cells. In the actin polymerization inhibit assay, A549 cells treated with cytochalasin D at a working concentration of 500 nM and then labeled F-Actin and the nuclei with the live cell fluorogenic F-actin-labeling probes SiR-actin (CY-SC001; Cytoskeleton, Inc.; red) and Hoechst 33342 (blue), respectively.

To investigate the functions of F-actin on adhesion and invasion of *M. catarrhalis*, A549 cells were incubated with different concentrations of cytochalasin D (100 nM, 500 nM, 1 μM, 5 μM, and 10 μM) in 0.2% DMSO for 45 min before infection with *M. catarrhalis* 73-OR isolate. Control wells contained only 0.2% DMSO. Gentamicin (300 μg/mL) was added to the infected monolayer cells before lysing with 0.25% trypsin and 0.1% saponin to kill any remaining adherent extracellular bacteria. The samples were plated on blood agar plates containing 5% sheep blood to determine the number of adherent or invasion bacteria. All assays were performed in six replicates. At least three independent experiments were performed.

### 2.7. Statistical Analysis

The bacterial burden between the 35-OR- and 73-OR-infected groups was compared using a two-tailed Mann–Whitney U-test. Pulmonary clearance and WBC count in mice belonging to the CO, 73-OR-, and 35-OR-infected groups were compared using a one-way analysis of variance (ANOVA). Western blot results were analyzed using ImageJ 1.53a (National Institutes of Health, Bethesda, MD, USA). Differences were considered statistically significant at *p* < 0.05.

## 3. Results

### 3.1. Murine COPD and Pulmonary Clearance Models

To explore the difference in pathogenicity in patients with COPD infected with *M. catarrhalis* with different phenotypes, we constructed a COPD mouse model and compared the pulmonary clearance rate in COPD mice infected with *M. catarrhalis* 35-OR (T35 group) or 73-OR (T73 group). Lung tissue injury in mice infected with 35-OR seemed to be more severe than that in mice infected with 73-OR (Figure 1A). More specifically, the alveolar septum collapsed, the alveolar cavity expanded, most alveolar epithelia sloughed off, and type II epithelial hyperplasia occurred in a few alveoli. Moreover, the alveolar interstitium of mice infected with 35-OR contained proliferating fibroblasts and epithelioid cells, as well as infiltrating lymphocytes.

The bacterial burden in the lungs of the T35 group was also significantly higher than that of the T73 group (*p* = 0.0104, Figure 1B). However, the peripheral blood WBC counts were not significantly different between the groups (Figure 1C).

### 3.2. Genome-Wide Gene Expression Analysis

The main difference between COPD mice infected with different phenotypic isolates was the pulmonary clearance rate. We then performed transcriptome sequencing of lung epithelial A549 cells infected with different phenotypic isolates. Transcriptome data were processed based on upregulated and downregulated differentially expressed genes (DEGs). A false discovery rate (FDR) of <0.05 and |log_2_Fold Change| of ≥0 were set as the cutoffs for DEG selection. Venn mapping showed 561 common DEGs between the T73 and CO groups and between T35 and CO groups (Figure 2Aa). In the comparison between the T73 and CO groups, 986 DEGs were identified (Figure 2Ab); 1196 DEGs were detected in the T35 and CO group comparison (Figure 2Ac). An additional 148 DEGs were identified between the T73 and T35 groups, of which 90 DEGs were upregulated, and 58 were downregulated in the T73 group compared with those in the T35 group (Figure 2Ad). The heatCluster graph indicated no complete separation between the T73 and T35 groups, but clustering could be observed (Figure 2Ae).

### 3.3. Functional Enrichment Analysis of DEGs

The GO, KEGG, and Reactome Pathway databases were employed to functionally classify the identified DEGs. In the GO analysis, the DEGs that were upregulated in the T73 group compared with those in the T35 group displayed enrichment in molecular functions that are associated with the regulation of actin cytoskeleton, including activities such as transcriptional activation and binding of myosin (Table 1).

Some cancer-related genes in A549 human lung adenocarcinoma epithelial cells are highly expressed [20]. In addition to background cancer pathways associated with A549 cells, the KEGG analysis revealed that most enriched pathways of the upregulated DEGs in the infected and CO groups were associated with the inflammatory response (Figure 2B). The top five pathways associated with the acute stage of infection were cytokine–cytokine receptor interaction, TNF signaling pathway, phosphatidylinositol-4,5-bisphosphate 3-kinase-AKT (PI3K-AKT) signaling pathway, mitogen-activated protein kinase (MAPK), and NOD-like receptor signaling.

### 3.4. Actin Cytoskeleton Affects Lung Epithelial Cell Infection with M. catarrhalis

The GO term enrichment analysis revealed that the actin cytoskeleton might be associated with the differences in the ability of 73-OR and 35-OR to infect A549 cells. *Moraxella catarrhalis* cells can adhere to or infect epithelial cells via a macropinocytosis-like mechanism involving localized F-actin polymerization [15]. During actin polymerization, the process of linking polymers of G-actin monomers to complex branched F-actin filaments occurs [21]. To investigate the functions of F-actin on the adhesion and invasion of *M. catarrhalis*, we used cytochalasin D to alter the polymerization kinetics of actin cytoskeleton filaments by irreversibly inhibiting the polymerization of G-actin to F-actin (Figure 3A). Although the inhibition of F-actin polymerization did not affect the ability of *M. catarrhalis* to adhere to A549 cells, it significantly inhibited the invasion of the cells (Figure 3B). Hence, *M. catarrhalis* cannot readily invade epithelial cells when actin polymerization is inhibited.

In addition, the F-actin/G-actin ratio exhibited a downward trend after infection; this decline was more evident in the T35 group (Figure 3C). In contrast, the G-actin level was not significantly affected by infection. Moreover, electron microscopy images of A549 cells co-cultured with *M. catarrhalis* revealed that the 73-OR was more likely to invade A549 cells (Figure 3Dii,Diii), whereas the 35-OR was more likely to adhere to the cell surface (Figure 3Div,Dv). Additionally, abundant microfilaments were commonly observed in the T73 group, whereas microfilaments were rarely observed in the T35 group. Numerous microfilaments were also observed in normal epithelial cells to maintain cell shape (Figure 3Di).

Furthermore, the fluorescence intensity of F-actin in lung epithelial cells infected with *M. catarrhalis* seems to have a tendency to decrease (Figure 3Ea). The length of F-actin in A549 cells did not change significantly after infection with *M. catarrhalis* (*p* < 0.05; Figure 3Eb). The fluorescence intensity of F-actin in the T35 group was significantly lower than that in the CO group (*p* = 0.0456) but not in the T73 group (*p* = 0.9824).

## 4. Discussion

The multidrug resistance of *M. catarrhalis* has increased gradually, especially to β-lactam and macrolide combinations [11,13,14]. In preliminary studies, it has been observed that *M. catarrhalis* isolates with distinct minimum inhibition concentration values exhibit varying degrees of adhesion and invasion capabilities on epithelial cells [14]. Although the interactions between epithelial cells and *M. catarrhalis* are critical for COPD pathogenesis [17], to the best of our knowledge, no study has evaluated the interaction between pulmonary epithelial cells and *M. catarrhalis* with different phenotypes.

In this study, we observed varying degrees of lung tissue damage in COPD mice induced by isolates with different phenotypes. This finding underscores the significance of investigating the genetic polymorphism of *M. catarrhalis* in future research. We also found that the pulmonary clearance rate of 73-OR (macrolide-susceptible and weak adhesion) was significantly higher than that of 35-OR (macrolide-resistant and strong adhesion) in the COPD murine model. This phenomenon is consistent with the findings of our previous study, where *M. catarrhalis* with different levels of macrolide susceptibility infected non-COPD mice [14].

From the host perspective, *M. catarrhalis* primarily inflicts damage on pulmonary epithelial cells to disrupt lung tissue. Through in vitro co-culture experiments, we sought to elucidate the alterations in human lung epithelial cells following infection with distinct phenotypic isolates. In our study, *M. catarrhalis* activated all TLR receptors and highly expressed NOD1/2 receptors in human epithelial cells, and no significant differences were observed between the T35 and T73 groups. The downstream nuclear factor-kappa B signaling pathway is also activated, mediating the expression of inflammatory factors. It has been shown that *M. catarrhalis* invasion is dependent on cellular microfilaments as well as bacterial viability and is characterized by macropinocytosis, leading to the formation of lamellipodia and engulfment of the invading organism into macropinosomes via recognition of the cell surface molecule TLR2 and the intracellular surveillance molecule NOD1 [15]. This process involves microfilament-dependent uptake, which requires signaling by the Rho family of small GTPases and reorganization of actin filaments [1]. Meanwhile, TLRs are also essential pattern recognition receptors of innate immunity, linking innate and adaptive immune responses [22].

The results also indicated statistically significant differences in myosin-related transcription factors within the epithelial cells infected with different isolates, as presented in Table 1. The cellular actin cytoskeleton consistently undergoes dynamic polymerization and depolymerization [23]. A recent study found that respiratory syncytial virus disrupts the airway epithelial barrier by decreasing cortactin levels and destabilizing F-actin [23]. In this study, we found that this mechanism occurred when lung epithelial cells were infected with *M. catarrhalis*. That is, F-actin polymerization and expression are reduced when *M. catarrhalis* infects lung epithelial cells, especially 35-OR (Figure 3C). At the same time, our study also confirmed that the polymerization and degradation processes of F-actin are not involved in the process of *M. catarrhalis* adhesion to lung epithelial cells (Figure 3B).

*Moraxella catarrhalis* utilizes the association of the ubiquitous surface protein A1 (UspA1) with integrin a5β1 to induce cytoskeletal rearrangement and ingestion of bacteria [1]. Although vitronectin/integrin-dependent cell adhesion and the subsequent internalization have been shown to contribute to the pathogenesis of specific respiratory pathogens, such as *Neisseria meningitidis* and *Streptococcus pneumoniae* [24], this mechanism has not been described for *M. catarrhalis*. Hence, the involvement of this mechanism in the infection of respiratory epithelial cells by *M. catarrhalis* with different phenotypes requires further investigation.

This study had some limitations. The sample size used in this study is limited. While bioinformatic analysis and in vitro phenotypic testing suggested that *M. catarrhalis* 35-OR and 73-OR are representative, future studies should include more isolates to confirm the generalizability of the findings. Additionally, we did not identify any specific genes that regulate the infection of lung epithelial cells by various phenotypic isolates. Therefore, we intend to incorporate selective inhibition of target genes as a potential enhancement to our RNA sequencing and data analysis methodology.

In conclusion, we compared two phenotypically distinct isolates of *M. catarrhalis* that infect COPD animals and interact with human lung epithelial cells. The results show that *M. catarrhalis* with different phenotypes exhibits different lung clearance rates in COPD mice; the lung clearance rate is related to the ability of pathogenic bacteria to adhere to lung epithelial cells. Moreover, the actin cytoskeleton was found to participate in *M. catarrhalis* invasion of epithelial cells, and inhibiting F-actin polymerization impairs internalization. This study provides a rationale for future immunotherapy against *M. catarrhalis* infection in COPD patients.

## Figures and Tables

**Figure 1 microorganisms-12-00291-f001:**
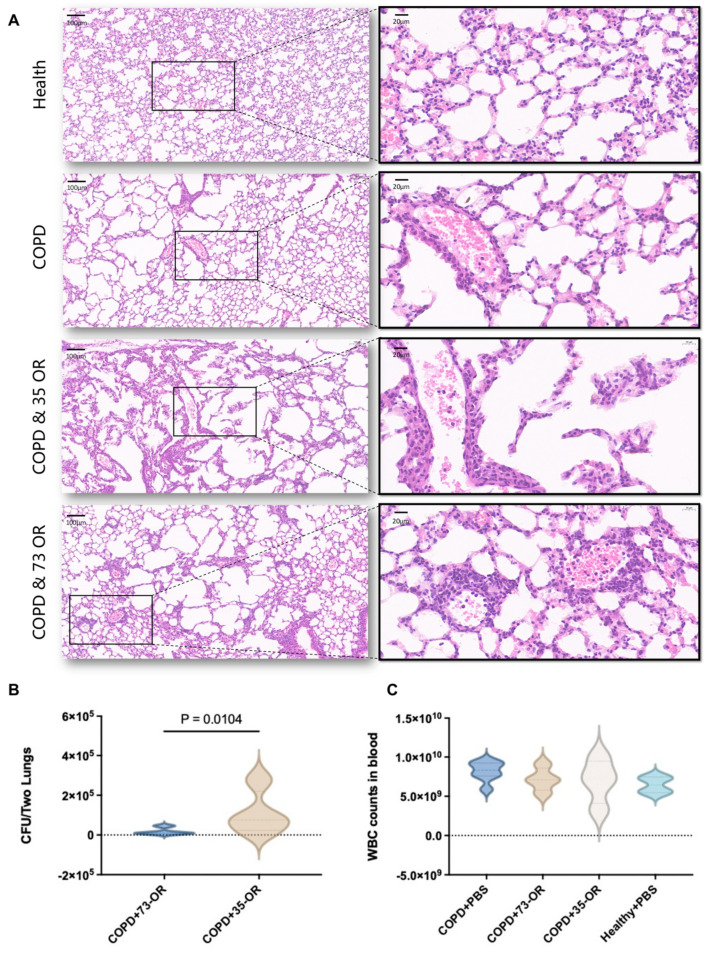
Murine chronic obstructive pulmonary disease (COPD) and pulmonary clearance models. (**A**) Histopathology of lung lesions in *C57BL/6J* mice. Hematoxylin and eosin (H&E)-stained pulmonary tissue at 3 h post-infection. Local magnification of blood vessels and the surrounding tissues revealed immune cells and peripheral inflammatory reactions. (**B**) Bacterial burden in the lungs. A significant difference (*p* = 0.0104) was observed in the colony forming unit (CFU) of *Moraxella catarrhalis* in the lung tissue infected with different phenotypes. (**C**). There was no significant difference in peripheral blood white blood cell (WBC) count after infection with different phenotypes of *M. catarrhalis*.

**Figure 2 microorganisms-12-00291-f002:**
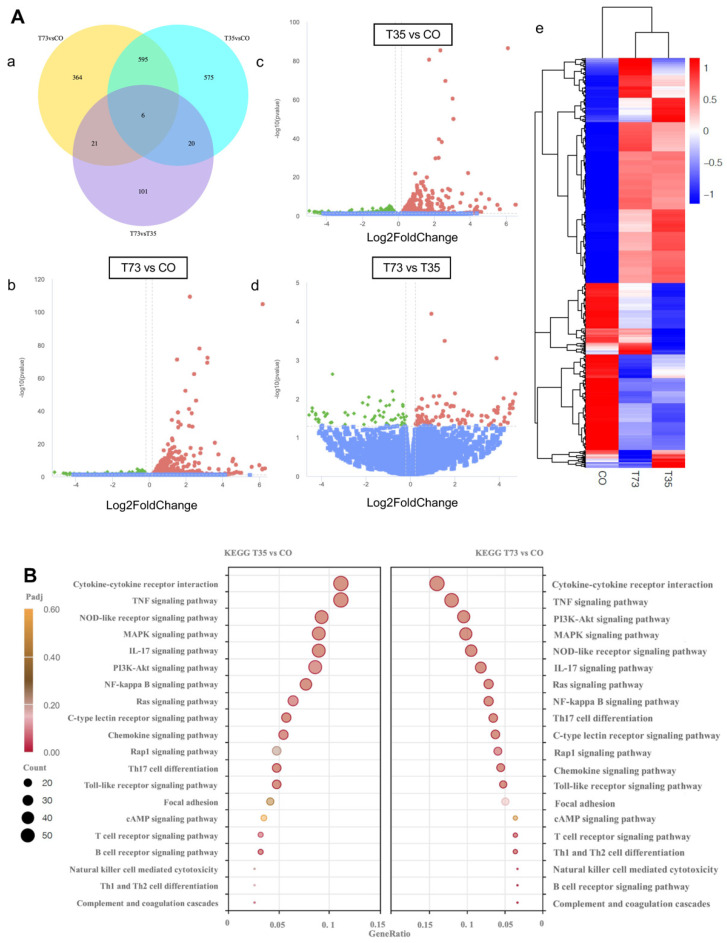
Interactions between *M. catarrhalis* and A549 cells explored using RNA sequencing. (**A**) Differential expression analysis. (**a**) Venn mapping depicting differentially expressed genes (DEGs) in different groups. (**b**–**d**) Volcano plot presenting the upregulated and downregulated DEGs in the infected group compared with those in the control group (red: upregulated; green: downregulated; blue: no significant change). (**e**) The heatCluster graph of DEG samples between different groups. (**B**) Kyoto Encyclopedia of Genes and Genomes (KEGG) pathway enrichment between the infected and control groups. A549 is a non-small-cell lung cancer cell line. Data analysis excluded A549 cell line background pathways, such as cancer pathways, Kaposi sarcoma-associated herpesvirus infection, and AGE-RAGE signaling pathway in diabetic complications.

**Figure 3 microorganisms-12-00291-f003:**
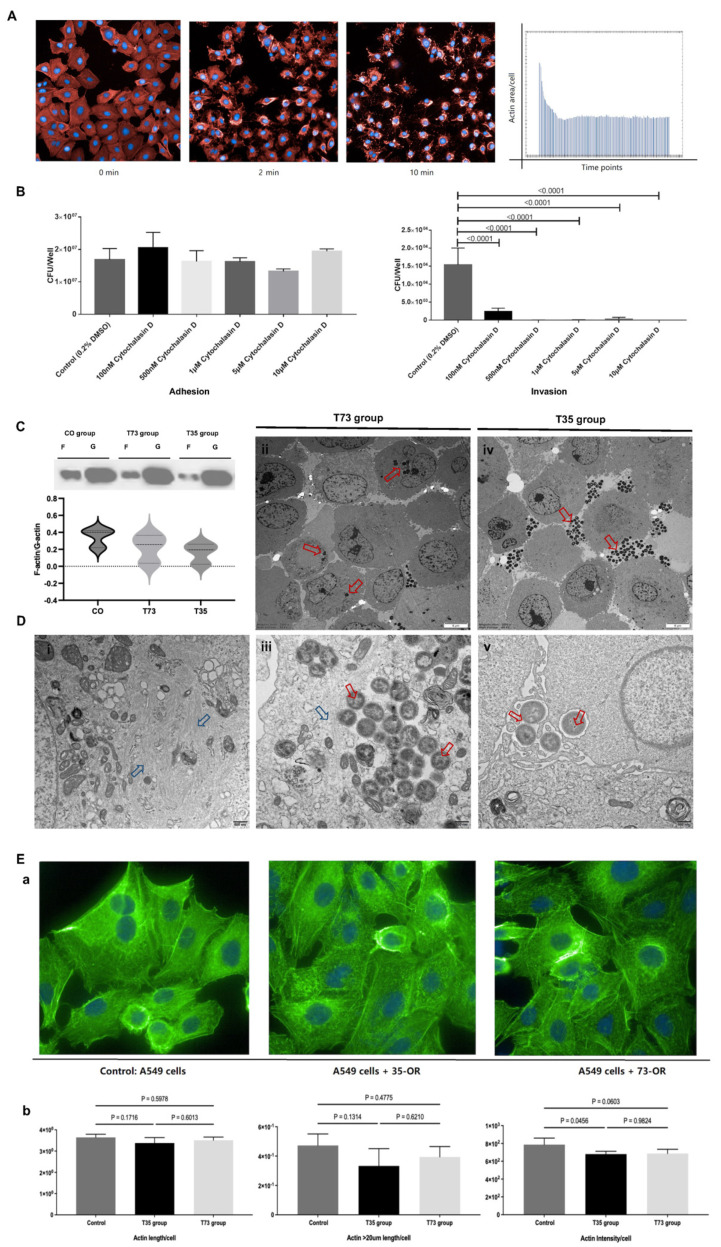
Inhibiting F-actin polymerization impairs the internalization of *Moraxella catarrhalis*. (**A**) Trend diagram of decrease in F-actin fluorescence area caused by cytochalasin D irreversibly inhibited F-actin polymerization in A549 cells (magnification: ×200). The F-actin and nuclei were labeled with SiR-actin (red) and Hoechst 33342 (blue), respectively. (**B**) Effect of cytochalasin D on the adhesion and invasion of *M. catarrhalis* with A549 epithelial cells. (**C**) Analysis of globular actin [G] and filamentous actin [F] content in the lung epithelial cells during *M. catarrhalis* infection. (**D**) Transmission electron microscopy analysis of *M. catarrhalis* adherence and invasion of A549 cells. (**i**) Morphological characteristics of A549 cells without infection, 73-OR (**ii**,**iii**) and 35-OR (**iv**,**v**) adhering to and invading A549 cells. Red arrows indicate *M. catarrhalis*; blue arrows indicate microfilaments. Scale bar: 5 μm (**ii**,**iv**) or 500 nm (**i**,**iii**,**v**). (**E**) Comparison of the morphology and expression intensity of F-actin in A549 cells using high-throughput imaging system (Operetta CLS; PerkinElmer, Waltham, MA, USA) and quantified using dedicated image analysis software (Harmony 4.9; PerkinElmer). Blue, nuclei; green, F-actin ((**a**), magnification: ×600). Quantitative analysis based on actin length and actin fluorescence intensity (**b**).

**Table 1 microorganisms-12-00291-t001:** Molecular function of Gene Ontology (GO) enrichment between the T73 and T35 groups, *p* adj < 0.05.

Category	GO ID	Description	Gene Ratio	BgRatio	*p*-Value	*p* adj	Gene Names
Molecular function	GO:0001228	Transcriptional activator activity, RNA polymerase II transcription regulatory region sequence-specific DNA binding	7/47	356/14430	0.0001	0.0186	*NFATC2*/*KLF7*/*MAFF*/*FOSB*/*IRF1*/*CSRNP1*/*EGR1*
Molecular function	GO:0032036	Myosin heavy chain binding	2/47	10/14430	0.0004	0.0303	*SPTBN5*/*CORO1A*
Molecular function	GO:0017022	Myosin binding	3/47	59/14430	0.0009	0.0370	*SPTBN5*/*NPC1L1*/*CORO1A*
Molecular function	GO:0001077	Transcriptional activator activity, RNA polymerase II proximal promoter sequence-specific DNA binding	5/47	243/14430	0.0011	0.0370	*NFATC2*/*KLF7*/*FOSB*/*IRF1*/*EGR1*

## Data Availability

The raw sequence data reported in this paper have been deposited in the Genome Sequence Archive (Genomics, Proteomics & Bioinformatics 2021) in the National Genomics Data Center (Nucleic Acids Res 2022), China National Center for Bioinformation/Beijing Institute of Genomics, Chinese Academy of Sciences (GSA-Human: HRA004725) that are publicly accessible at https://ngdc.cncb.ac.cn/gsa-human.

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
