# Peer review of "Inhibiting F-Actin Polymerization Impairs the Internalization of Moraxella catarrhalis"

_microorganisms, 2024, doi:10.3390/microorganisms12020291_

Round 1

Reviewer 1 Report

Comments and Suggestions for Authors

The manuscript is interesting and open to further investigation on this topic. Some minor comments can be done. The  "RNA sequencing and data analysis" could be improved by using selective inhibition of target genes. Alternatively, a more consistent explanation of this step should be included in the Discussion. The role of the tissue's  inflammatory response, if any, should be specifically evaluated or at least discussed as a potential interference and limitation. In the Introduction the aim should be better presented and if more aims are proposed they should be presented clearly and explained avoiding the redundancies with previous publications from the same group. In the Discussion the importance/utility of the results should be better discussed avoiding speculative consideration.

Comments on the Quality of English Language

Small typing errors should be revised.

Author Response

Dear Reviewer,

Thank you for your thorough review, insightful comments, and constructive suggestions, all of which have considerably enhanced the presentation of our manuscript. We have meticulously revised the manuscript in line with your remarks.

We trust that our revision has adequately addressed all the issues raised, and we are hopeful that our manuscript is now suitable for publication. 

Response to Reviewer

Comment 1:

The "RNA sequencing and data analysis" could be improved by using selective inhibition of target genes. Alternatively, a more consistent explanation of this step should be included in the Discussion.

Response: We sincerely appreciate your insightful comments, which undoubtedly contribute to the refinement of our research.

We have carefully reviewed your comments, particularly those concerning the "RNA sequencing and data analysis" section. We acknowledge the significance of enhancing this aspect to improve the clarity and overall quality of our work. Due to current limitations in our laboratory conditions, we plan to address this concern in our future work. We intend to incorporate selective inhibition of target genes as a potential enhancement to our RNA sequencing and data analysis methodology. It's important to note that we will explicitly mention these laboratory constraints in the "Limitations" section to provide transparency.

Moreover, we understand the importance of a more consistent explanation of this step in the Discussion section. For specific modifications, please refer to our latest manuscript submission. Once again, we appreciate your thoughtful input and look forward to further refining our research based on your suggestions.

Comment 2:

The role of the tissue's inflammatory response, if any, should be specifically evaluated or at least discussed as a potential interference and limitation.

Response: We appreciate the time and effort you have dedicated to the review process.

The purpose of the histopathological examination was to assess the differential virulence of different phenotypic strains. We found that the genes of different phenotypic clones exhibited polymorphism. In vitro experiments also demonstrated differences in the invasive or adhesive abilities of the isolates 35-OR and 73-OR to pulmonary epithelial cells. Through in vivo animal experiments, we aim to further confirm these findings. Based on our research results, it is evident that different phenotypic strains cause varying degrees of lung tissue damage. In future studies, we will continue to investigate the molecular mechanisms responsible for these differences.

Comment 3:

In the Introduction the aim should be better presented and if more aims are proposed they should be presented clearly and explained avoiding the redundancies with previous publications from the same group. In the Discussion the importance/utility of the results should be better discussed avoiding speculative consideration.

Response: Thank you for your insightful writing suggestion; It have been incredibly beneficial.

We have thoroughly reviewed and revised the manuscript, clarifying any ambiguous descriptions and correcting the inaccuracies. In response to the introduction, we have provided additional background information focused on the objectives of our study. In the discussion section, we have taken your suggestions into consideration and provided an in-depth discussion based on our research findings. Please refer to our latest manuscript submission for specific modifications.

Reviewer 2 Report

Comments and Suggestions for Authors

The manuscript of Yu et al., shows some experiments to elucidate the factors responsible for the pathogenicity of Moraxella catarrhalis in exacerbation of chronic obstructive pulmonary disease. They performed a comparison between two phenotypically distinct strains of M. catarrhalis and observed that different phenotypes exhibit different lung clearance rates in COPD mice, based on the ability of pathogenic bacteria to adhere to lung epithelial cells. Moreover, they show that inhibition of F-actin polymerization impaired M. catarrhalis internalization, a mechanism that could be used for individualized clinical treatment of patients with AECOPD infected with M. catarrhalis. So, the article „Inhibiting F-actin polymerization impairs the internalization of Moraxella catarrhalis” can be published in Microorganism.

Author Response

Dear Reviewer,

We sincerely appreciate your constructive feedback and positive evaluation of our manuscript titled "Inhibiting F-actin polymerization impairs the internalization of Moraxella catarrhalis." Your insightful comments have been instrumental in refining our study.

  1. Comparison of M. catarrhalis Phenotypes: You correctly highlighted the significance of our comparison between two phenotypically distinct strains of M. catarrhalis. We agree that understanding the differences in lung clearance rates based on adherence to lung epithelial cells is crucial for unraveling the pathogenicity of this bacterium in COPD mice.

  2. Implications for Individualized Clinical Treatment: Your recognition of the potential clinical implications of our findings, particularly regarding the individualized treatment of patients with AECOPD infected with M. catarrhalis, aligns with our research objectives. We are pleased that you acknowledge the relevance of our study to clinical applications.

We would like to express our gratitude for your thorough review and valuable suggestions, which have undoubtedly enhanced the quality and clarity of our manuscript. 

Thank you for considering our manuscript for publication in Microorganisms.

Sincerely,

Jinhan